# Proximity interactome of alphavirus replicase component nsP3 includes proviral host factors eIF4G and AHNAK

Aditya Thiruvaiyaru[1], Sari Mattila[1], Mohammadreza Sadeghi[1], Krystyna Naumenko[2], Andres Merits[2], Markku Varjosalo[3], Tero Ahola[1]*

1 Department of Microbiology, Faculty of Agriculture and Forestry, University of Helsinki, Helsinki, Finland,
2 Institute of Bioengineering, University of Tartu, Tartu, Estonia, 3 Institute of Biotechnology, HiLIFE Helsinki Institute of Life Science, University of Helsinki, Helsinki, Finland

* tero.ahola@helsinki.fi

## Abstract

All positive-strand RNA viruses replicate their genomes in association with modified intracellular membranes, inducing either membrane invaginations termed spherules, or double-membrane vesicles. Alphaviruses encode four non-structural proteins nsP1-nsP4, all of which are essential for RNA replication and spherule formation. To understand the host factors associated with the replication complex, we fused the efficient biotin ligase miniTurbo with Semliki Forest virus (SFV) nsP3, which is located on the cytoplasmic surface of the spherules. We characterized the proximal proteome of nsP3 in three cell lines, including cells unable to form stress granules, and identified >300 host proteins constituting the microenvironment of nsP3. These included all the nsPs, as well as several previously characterized nsP3 binding proteins. However, the majority of the identified interactors had no previously identified roles in alphavirus replication, including 39 of the top 50 interacting proteins. The most prominent biological processes involving the proximal proteins were nucleic acid metabolism, translational regulation, cytoskeletal rearrangement and membrane remodeling. siRNA silencing confirmed six novel proviral factors, USP10, AHNAK, eIF4G1, SH3GL1, XAB2 and ANKRD17, which are associated with distinct cellular functions. All of these except SH3GL1 were also important for the replication of chikungunya virus. We discovered that the small molecule 4E1RCat, which inhibits the interaction between the canonical translation initiation factors eIF4G and eIF4E, exhibits antiviral activity against SFV. Since the same molecule was previously found to inhibit coronaviruses, this suggest the possibility that translation initiation factors could be considered as targets for broadly acting antivirals.

## Author Summary

Alphaviruses are arthropod-borne positive-strand RNA viruses causing febrile diseases in many parts of the world. Since RNA viruses encode a very limited number of proteins, they co-opt multiple cellular proteins and pathways to assist in virus replication.

**Data availability statement:** The mass spectrometry data generated in this study have been deposited in the MassIVE database (massive.ucsd.edu) under the accession number MSV000096380. All the other data that support the findings of this study are available within the paper and its Supplementary Information and Supplementary Data.

**Funding:** This work was funded by the Finnish Cultural Foundation (https://skr.fi/en) by the Jane and Aatos Erkko Foundation (https://jaes.fi/en/frontpage/) and the Academy of Finland (https://www.aka.fi/en/) (grant number 361249), all to T.A. The University of Helsinki Research Foundation (https://www.helsinki.fi/en/about-us/strategy-economy-and-quality/university-finances/university-helsinki-research-found-tion) provided the salary funding of A.T. S.M. was funded by the Academy of Finland (https://www.aka.fi/en/) postdoctoral researcher grant (number 333276), and M.V. by the Academy of Finland (https://www.aka.fi/en/) (grant number 336470). K.N. and A.M. were funded by the Estonian Research Council (https://etag.ee/en/) (grant number PRG1154). Open access funded by Helsinki University Library. The funders had no role in study design, data collection and analysis, decision to publish, or preparation of the manuscript.

**Competing interests:** The authors have declared that no competing interests exist.

To better understand the host proteins involved in alphavirus replication, we have identified proteins that are located in the vicinity of alphavirus replication protein nsP3. nsP3 is essential for alphavirus RNA replication, and it also disassembles cellular stress granules that are induced during infection. Therefore, we also characterized the nsP3 interacting proteins in cells unable to form stress granules, eliminating stress granule proteins from the overall dataset. nsP3-interacting proteins were often involved in cellular translation, RNA metabolism, membrane remodeling and cytoskeletal rearrangement. Gene silencing analysis identified six novel proviral proteins for alphaviruses. Targeting one of these, the canonical translation initiation factor eIF4G, with a small molecule inhibitor, reduced virus replication. This work provides a starting point for comprehensive analysis of critical alphavirus host factors interactions, which can reveal new targets for broadly acting antivirals.

## Introduction

Alphaviruses are arthropod-borne viruses causing febrile disease with arthritis or encephalitis. The most widespread alphavirus is the arthritogenic chikungunya virus (CHIKV), found throughout the tropics and subtropics, and spreading towards temperate areas with the mosquito vector *Aedes albopictus* [1]. Significant regional pathogens include Sindbis virus (SINV) in Northern Europe and Ross River virus in Australia, Venezuelan equine encephalitis virus (VEEV) and Mayaro virus in the Americas as well as o'nyong'nyong virus in Africa [2–5].

The alphavirus genome is a positive-sense RNA of ~12 kb, which encodes four non-structural or replication proteins nsP1-nsP4, as well as the structural capsid and envelope proteins. The nsPs are produced as a polyprotein precursor that is sequentially cleaved to the four end products, which are all required for RNA replication [6]. Positive-strand RNA viruses induce alteration of cellular membranes to create protected environments for viral RNA replication [7–9]. Bulb-shaped membrane invaginations or spherules consist of a narrow neck region connecting the interior of the invagination to the cytoplasm. Spherules are induced by viruses of the phylum *Kitrinoviricota*, which includes alpha-, noda- and flaviviruses as well as many plant virus groups [10,11].

Recent high-resolution structural studies have indicated that nsP1, which is the RNA capping enzyme and membrane anchor of the alphavirus replicase, forms a dodecameric ring at the neck of the spherule [12,13]. At the center of this ring lies a single copy of the RNA-dependent RNA polymerase nsP4 [14], thus ideally placed to copy the RNA template primarily located in the spherule interior. Connected to nsP4 on the cytoplasmic side was found a single copy of the RNA-helicase/protease nsP2 [14]. Interestingly, further towards the cytoplasm outside this ~0.9 MDa complex of nsP1/2/4, there is an even larger protein complex of ~1.2 MDa [14,15]. Based on the tubular structure of large nsP3 multimers, this complex was proposed to consist mainly of nsP3 [16].

The specific functions of nsP3 in RNA replication remain uncertain. The N-terminus of nsP3 contains a macrodomain involved in removing ADP-ribosyl modifications from proteins [17], an activity crucial for virus replication and pathogenesis [18]. The central region of nsP3 structurally forms a zinc-binding domain involved in homo-multimerization [16,19], and the C-terminus contains an unstructured tail, variable in sequence between different alphaviruses. The tail region is known to interact with several host proteins [20]. Early studies of proteins binding to nsP3 identified the stress granule core protein G3BP (ras-GTPase-activating protein SH3-domain binding protein) as a major interacting protein [21,22]. The so-called Old-World alphaviruses (including SINV and CHIKV) utilize G3BPs to facilitate replication complex

assembly, whereas New World alphaviruses exploit fragile X syndrome (FXR) family proteins for the same purpose [23]. Another directly interacting protein important for virus replication and shared by most alphaviruses is the membrane-modifying protein amphiphysin [24]. Several other nsP3-interacting proteins have been discovered [20], including those important for individual viruses, such as FHL (four-and-a-half LIM domain) proteins utilized by CHIKV [25,26]. Besides the viral replication spherules at membranes, nsP3 is found in alternative cytoplasmic foci, where it diverts the G3BPs in breaking up the stress granules [27,28].

Although a few important host protein interactions have been recognized and even mapped in alphavirus nsP3, the roles of nsP3 in replication remain incompletely understood [20]. Here, we turned to proximity-dependent labeling to characterize the protein environment of nsP3 in detail within Semliki Forest virus (SFV)-infected cells. SFV appears to be less dependent on host factors than CHIKV or other alphaviruses, and specifically, it is not dependent on the stress granule inducing function of G3BPs [20,29]. This permitted the comparison of host interactions in the presence and absence of stress granules, to demonstrate the stress granule-independent nsP3-host interactions. We fused nsP3 with the biotin ligase miniTurbo (mTB) [30], which allows short labeling periods for efficient proximity detection. In cells infected with the fusion protein-expressing SFV, we detected multiple high-confidence host proteins, including the previously described nsP3 interactors and many novel proteins, which include the large scaffolding protein AHNAK, and factors involved in protein translation. AHNAK, eIF4G1, and several other proteins were found to be important for alphavirus replication.

## Results

### Generation of mutant SFV expressing highly active biotin ligase

To identify host proteins proximal to the alphavirus replicase component nsP3, we constructed a recombinant SFV by fusing an engineered biotin ligase mTB (~28 kDa) in-frame with nsP3 within the infectious cDNA clone, as schematically shown in S1A Fig. mTB was selected due to its small size and rapid biotin labeling activity, requiring only approximately 15 minutes of incubation with exogenously added biotin for detectable signals [30]. We first compared the relative levels of expression of mutant nsP3 with the wild-type protein. For this, the infectious cDNA clones were transfected into BHK-21 cells, which are highly permissive for SFV infection. At 18 h post transfection the expression level of nsP3-mTB was comparable to that of wild-type nsP3. Additionally, nsP1 and nsP4 levels were similar between the mutant and wild-type viruses (Fig 1A).

Next, we compared the replication efficiency of the mutant virus SFV4-nsP3-mTb with that of the wild-type SFV4. BHK-21 cells were infected at a multiplicity of infection (MOI) of 10, collected and lysed at the indicated time points (Fig 1B). Similar intensities of nsP3 and nsP3-mTB were observed on the Western blot, and stable expression of nsP3-mTb was confirmed. From a separate set of infected cells, supernatants were collected, and the viral titers were quantified (Fig 1C). Overall, little difference in titers was observed between SFV4 and SFV4-nsP3-mTb, indicating similar replication efficiencies.

To confirm the enzymatic activity of the biotin ligase in nsP3-mTB, BHK-21 cells infected with the mutant virus were supplemented with 50 μM exogenous biotin for periods ranging from 15 to 180 min. Biotinylated proteins were detected in cell lysates as early as 15 min post-biotin addition, with signal intensity increasing over time (Fig 1D). To detect the biotinylation at an early stage of infection, when replication complexes are forming [31], the cells were infected at the high MOI of 50 and supplemented with biotin for 15-30 min at 2.5 h post infection (Fig 1E). High multiplicity infection ensures the synchronous infection all the cells

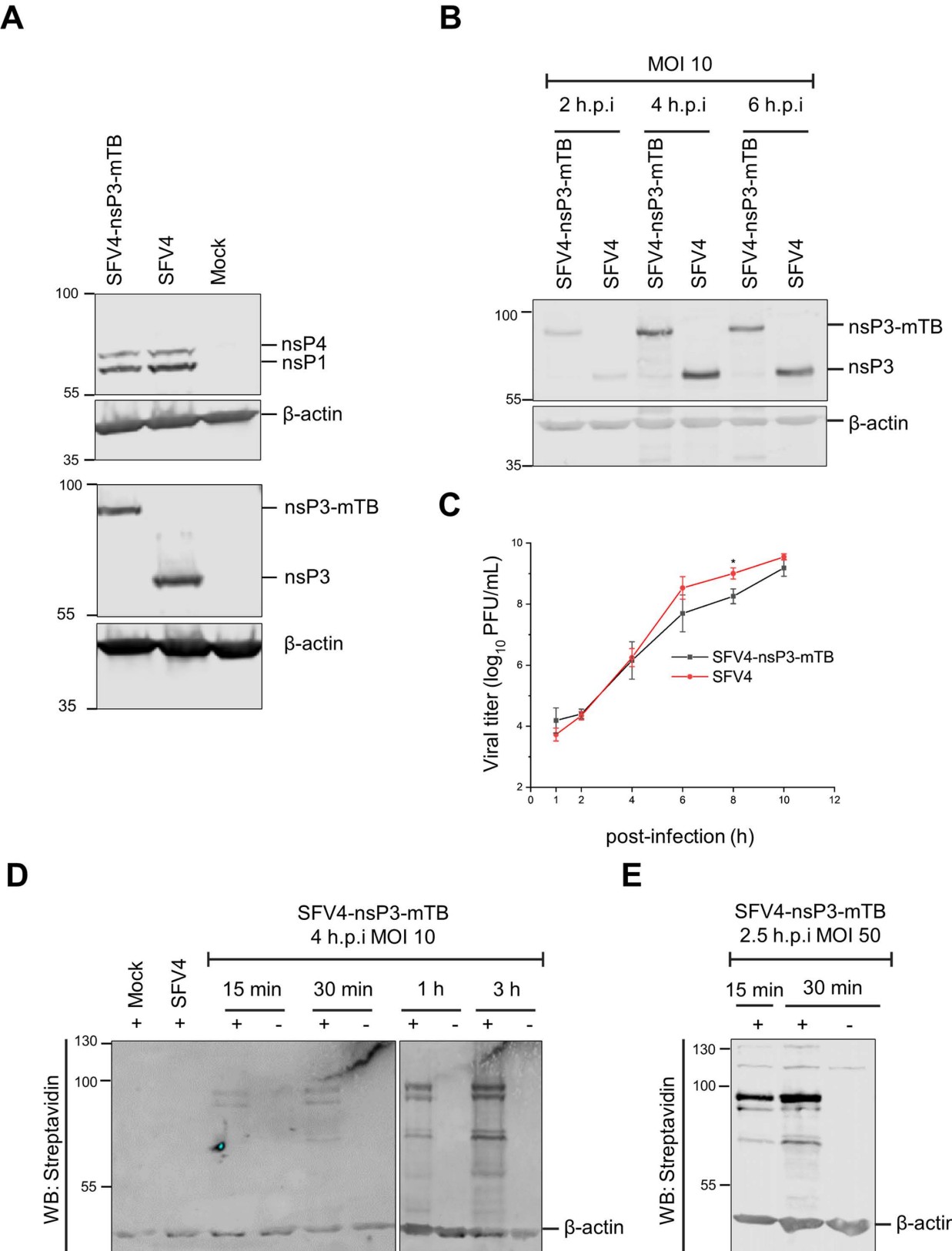

**Fig 1. Virus stability, replication efficiency and protein biotinylation during infection with SFV expressing nsP3-mTB. A)** BHK-21 cell lysates were collected 18 h post transfection with the indicated infectious constructs and analyzed by Western blotting with antibodies against

nsP1, nsP3, and nsP4; β-actin was used as a loading control. **B)** Expression levels of nsP3/nsP3-mTB in cell lysates collected at the indicated times post-infection, detected by Western blotting. **C)** Viral titers from growth media collected at the indicated times post-infection (MOI=10), measured by plaque assay. The graph is a mean of three independent biological replicates and the error bars indicate standard deviation. **D)** BHK-21 cells were infected with SFV4 or SFV4-nsP3-mTB at MOI 10. At 4 h.p.i. the cells were either exposed to 50 µM biotin (+) or not (-), for the time interval indicated. The mock infected and SFV4 infected samples were collected after exposure to 50 µM biotin for 15 min - 3 h. **E)** BHK-21 cells were infected at MOI 50, and at 2.5 h post infection the cells were exposed to 50 µM biotin for the indicated times. Samples were analyzed through Western blotting by detecting biotinylated proteins with streptavidin and β-actin as a loading control.

and efficient formation of replication complexes. Again, it was evident that the biotin ligase is enzymatically active, and biotinylation can be detected even after the shortest incubation time.

## SFV nsP3 interactome determined with proximity labelling

To elucidate the interactome of nsP3 at an early stage of infection, we selected the time point of 2.5 h in BHK-21 cells after infection at MOI 50 with either SFV4-nsP3-mTB or SFV4 as a control. To have a low background of biotinylation, cells were exposed to exogenous biotin for only 15 min [30]. The lysates were purified with streptavidin beads and first analyzed either through silver staining of protein gels or immunoblotting, and then processed for mass spectrometry. To compare the nsP3 interactome in a human cell context, we performed similar experiments in human osteosarcoma U2OS cells. These cells have been extensively used in studies of alphavirus infection and the roles of host factors. Furthermore, U2OS cell lines lacking the endogenous G3BPs 1 and 2 (ΔΔG3BP1/2) are defective in stress granule formation, and the reconstitution of G3BP1 and its truncated variants restores the different functions of G3BPs [29,32]. We used the double knock-out cell lines (ΔΔG3BP1/2) that were reconstituted either with full-length G3BP1 or with the short N-terminal NTF2-like domain of G3BP1, termed G3BP1(1-135), each fused with green fluorescent protein located at their N-terminus. These cell lines are referred to as GFP-G3BP1 and GFP-G3BP1(1-135), respectively. In the latter cells, stress granules are not restored, but SFV replicates to the same high levels observed in GFP-G3BP1 cells, whereas SFV replicates poorly in the complete absence of any G3BP domains (ΔΔG3BP1/2 cells)[29]. Thus, G3BP1(1-135) cells enable us to study the SFV nsP3 interactome in the absence of stress granules. In contrast, CHIKV is unable to replicate in GFP-G3BP1(1-135) cells [29]. The time points used for biotin labeling of the U2OS-derived cell lines were selected to obtain early time points of infection but already with a high-level expression of SFV nsP3 or nsP3-mTB (S1B Fig; see Materials and Methods).

We observed enrichment of biotinylated proteins in the fractions purified with streptavidin beads (Fig 2A, B and C). A prominent (doublet) band around ~90 kDa suggested the self-biotinylation of nsP3-mTB. This prominent band as well as several other bands were also observed in silver-stained gels, which indicated that multiple proteins not observed under control conditions had been enriched (Fig 2D, E and F). Comparing the patterns between GFP-G3BP1 and GFP-G3BP1(1-135) cells, we observed noticeable differences in the distribution of biotinylated and enriched proteins. The absence of proteins essential for stress granule formation in the GFP-G3BP1(1-135) cells may lead to an overall reduction in proteins visualized (compare Figs 2B vs. 2C and 2E vs. 2F).

To analyze the nsP3 interactome, we compared protein enrichment between cells infected with SFV4-nsP3-mTB and those infected with SFV4. Label-free quantification (MaxLFQ) intensities were normalized based on overall protein abundance. Proteins with an adjusted p-value < 0.05 and a $\log_2$ fold change >1 were considered high-confidence hits. Using these criteria, we identified 350 proteins in SFV4-nsP3-mTB infected BHK-21 cells that are likely proximal to and potentially interacting with SFV nsP3 (Fig 3A; S1 Table). The top 50 (by $\log_2$ fold change) hit proteins are described in detail in S2 Table. As expected, our dataset included known alphavirus

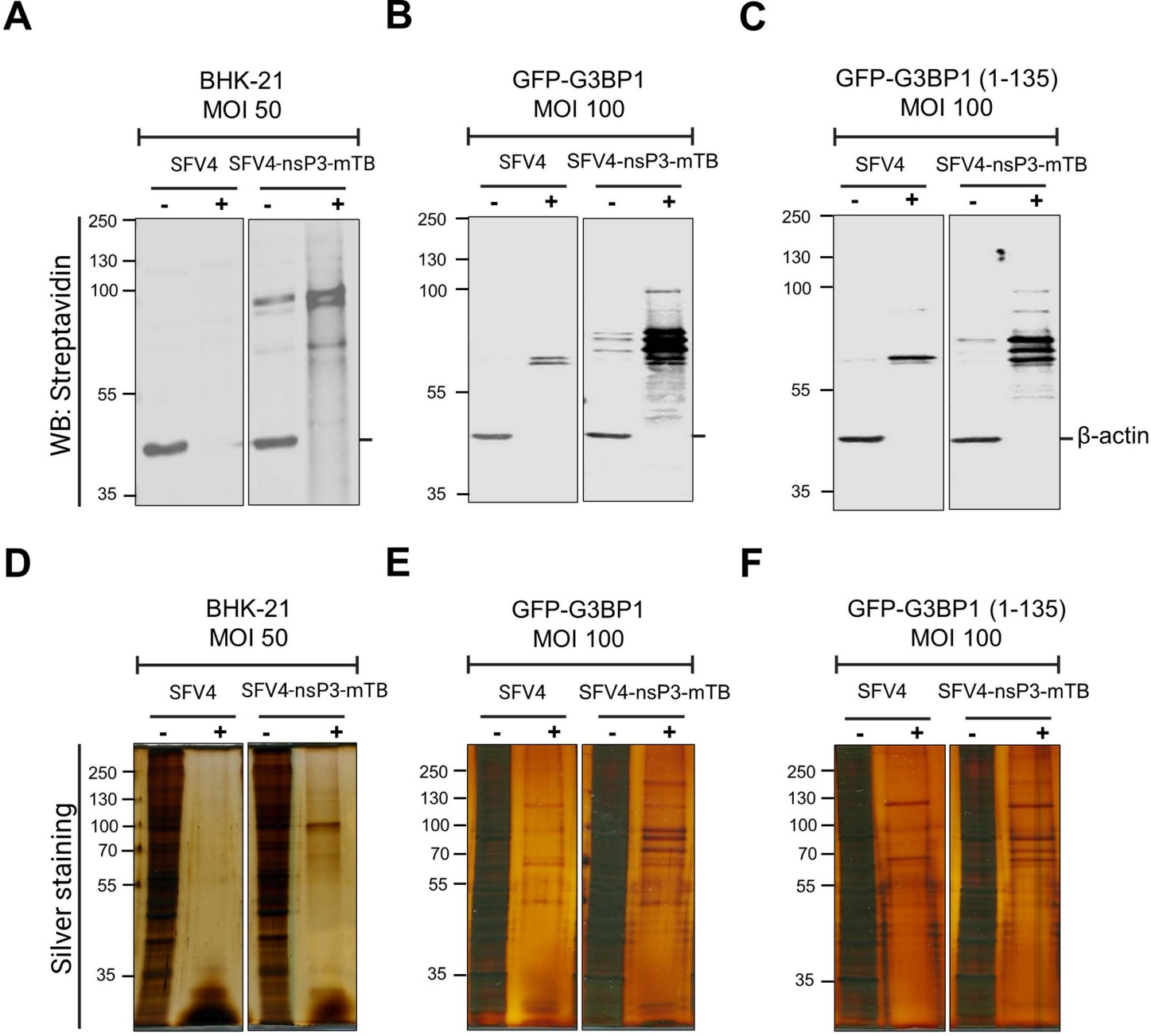

**Fig 2. Affinity purification of biotinylated proteins after proximity labelling in different cell lines. A-F).** The indicated cell lysates were purified using streptavidin coated magnetic beads (+). ~20% of the sample was collected and analyzed through Western blotting by detecting streptavidin and β-actin **(A, B & C)**. The same fractions were analyzed by staining with silver nitrate **(D, E & F)**. The pre-purified lysate (-) was used as loading control and to demonstrate the enrichment of biotinylated protein in the streptavidin purified fraction (+). The images show one representative of three independent biological replicates that were analyzed through mass spectrometry.

host factors such as amphiphysin-2 (BIN1), G3BP1, CD2-associated protein (CD2AP), SH3-domain kinase binding protein 1 (SH3KBP1), FHL1, zinc finger CCCH-type antiviral protein 1 (ZC3HAV1), and FXR1. All the non-structural viral proteins nsP1, nsP2, nsP3, and nsP4 were detected. Importantly, many new host factors not yet studied in the context of alphavirus infection were revealed: of the top 50 hits, 39 were novel interactors (S2 Table; Figs 3A S2).

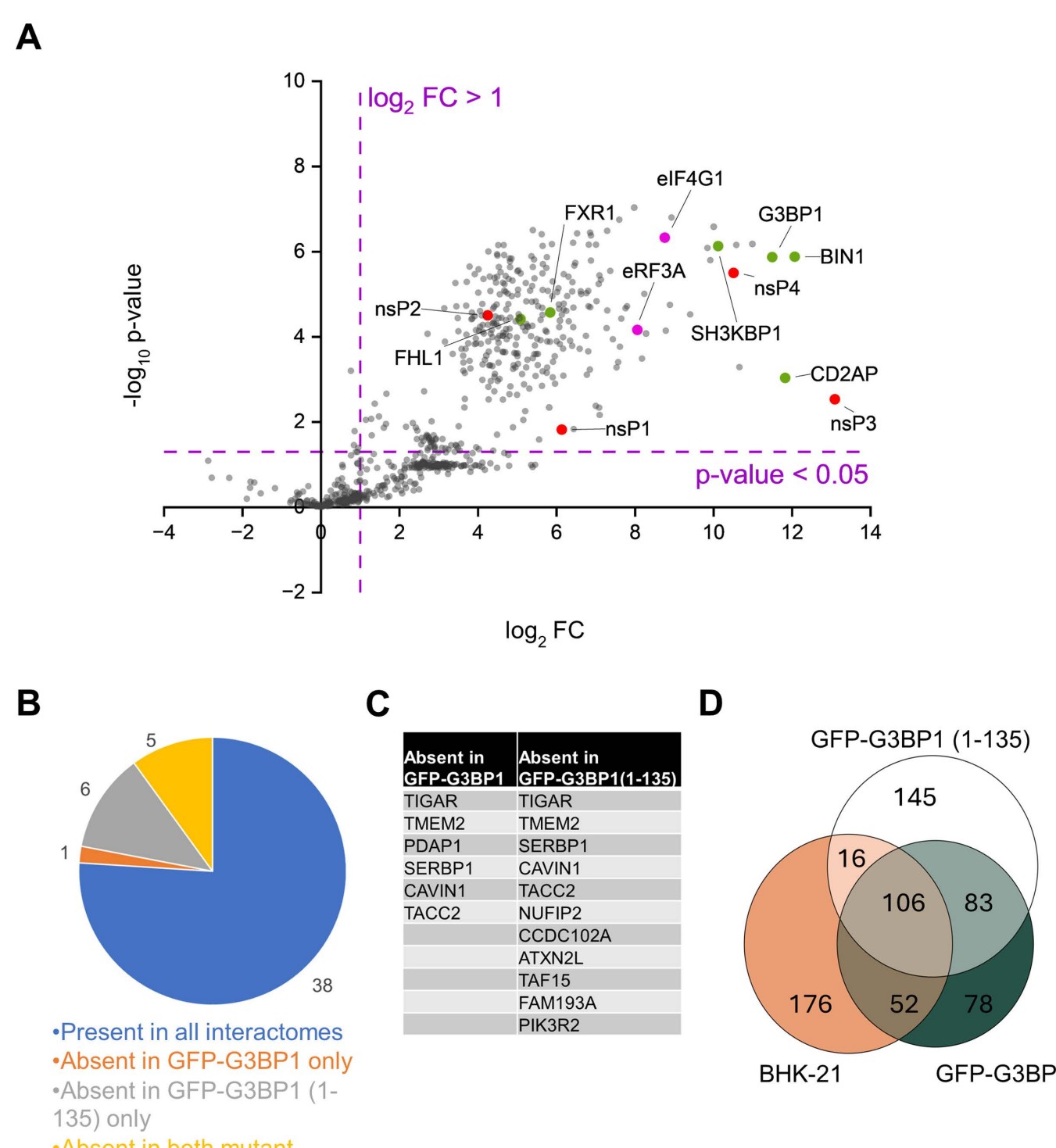

**Fig 3. Proteins detected in the SFV nsP3 interactome. A)** Volcano plot representing the distribution of the proteins detected in BHK-21 cells through proteomic analysis. High confidence interactors were determined as $\log_2$ Fold change > 1 and -$\log_{10}$ adjusted p-value >1.3 (i.e. p-value < 0.05; independent Student's t-test), indicated with the dotted lines. Viral non-structural proteins are marked with red, the most prominent previously known interactors are marked with green [20,24,43], and the two translation factors are indicated in magenta. The volcano plots for the mutant U2OS cell lines are represented similarly in S2 Fig **B) and C)** Distribution of the top 50 interactors, based on the BHK-21 dataset, in the U2OS-derived cell line datasets. **D)** Venn diagram representing the overlap of the interactome of nsP3 between the three cell lines: BHK-21, GFP-G3BP1 and GFP-G3BP1(1-135). The number of proteins within each sector are shown.

Most of the top hits identified in BHK-21 cells were also present in the datasets of SFV4-nsP3-mTB infected U2OS-derived cell lines, except for stress granule-associated proteins, which were absent in the GFP-G3BP1(1-135) cells (Fig 3B and 3C). A Venn diagram (Fig 3D) indicates the overlaps and differences between the proteins interacting with nsP3 in different cell lines, with 106 proteins common in all three cell lines. Among the identified 158 common proteins between GFP-G3BP1 and BHK-21 cell lines, the proteins ATXN2L, CAPRIN1, ED3, NUFIP2, RBM22, RC3H1, STAU1, TARDBP, YTHDF2, and YTHDF3 that have been previously associated with the assembly of stress granules [33,34], are entirely absent from the GFP-G3BP1(1-135) dataset. The few proteins associated with stress granules that are present in all three cell lines include UBAP2L and USP10, which interact with the NTF2-like domains present in the N-terminus of G3BP1; this accounts for their presence in all the cell lines [32].

Gene ontology analysis of the proteome highlighted biological processes predominantly related to nucleic acid metabolism, particularly RNA stability and catabolism (Fig 4; S3 Table). This includes factors that may assist in virus genome replication and subgenomic RNA transcription, such as the exoribonuclease XRN1, previously implicated as essential for SINV replication [35]. A significant portion of the RNA regulation cluster comprises the heterogeneous nuclear ribonucleoproteins (hnRNPs), which are involved in post-transcriptional processing and fate of RNAs. Besides the nucleus these proteins are present in the cytoplasm, which fraction could be increased during infection, as shown for several alphaviruses [36,37]. While nsP2 is known to localize to the nucleus and may facilitate interactions with nuclear proteins [38], the proximity of these host factors to nsP3-mTB suggests potential cytoplasmic interactions or redistribution during infection.

Other major categories identified include translational regulation, cytoskeletal rearrangement, and protein complex assembly. Translation-associated proteins, such as DEAD-box helicase 3 (DDX3X), have been implicated in the translation of New World alphaviruses, including Western equine encephalitis virus and VEEV [39,40]. Also, a significant portion of the translation initiation factor complex proteins including eIF2A, eIF4B, eIF4G1, eIF4G2, eIF4G3, and eIF5 [41] are found in the interactome, as are other translation factors, such as eEF2 and eRF3A.

The proteins CAPZA1 and CAPZB have an effect in the rearrangement of cytoskeleton through interactions with actin that could influence spherule formation or mobilization, and they are essential for the replication of alphaviruses [42,43]. The membrane remodeling amphiphysins have been previously shown to interact with nsP3 and are important for the replication of alphaviruses [24]. Additional SH3-domain containing proteins interacting with nsP3 include CD2AP and its homolog SH3KBP1 [43], also detected using our proximity labeling approach. Proteins involved in protein complex assembly, such as molecular chaperones including CCT8, may be essential for the formation of viral replication complexes or the assembly of cellular complexes involved in translation initiation and ribosome assembly. This is supported by gene ontology terms associated with viral protein synthesis and viral replication complex formation (Fig 4). Gene ontologies are defined based on the known physiological roles of the genes and gene products. During virus infections, proteins can be recruited to additional moonlighting roles, as seen by the translocation of some nuclear proteins to cytoplasm during alphavirus infection [44].

## Identification of pro- and antiviral factors in the nsP3 interactome

To assess the effect of the identified host factors on SFV replication, we used a custom siRNA library to transiently reduce the expression of each of the proteins. We selected the top 50 high-confidence hits from the BHK-21 interactome (S2 Table) and studied their effect on SFV infection in U2OS cells. At 3 days post siRNA-transfection, the cells were infected with

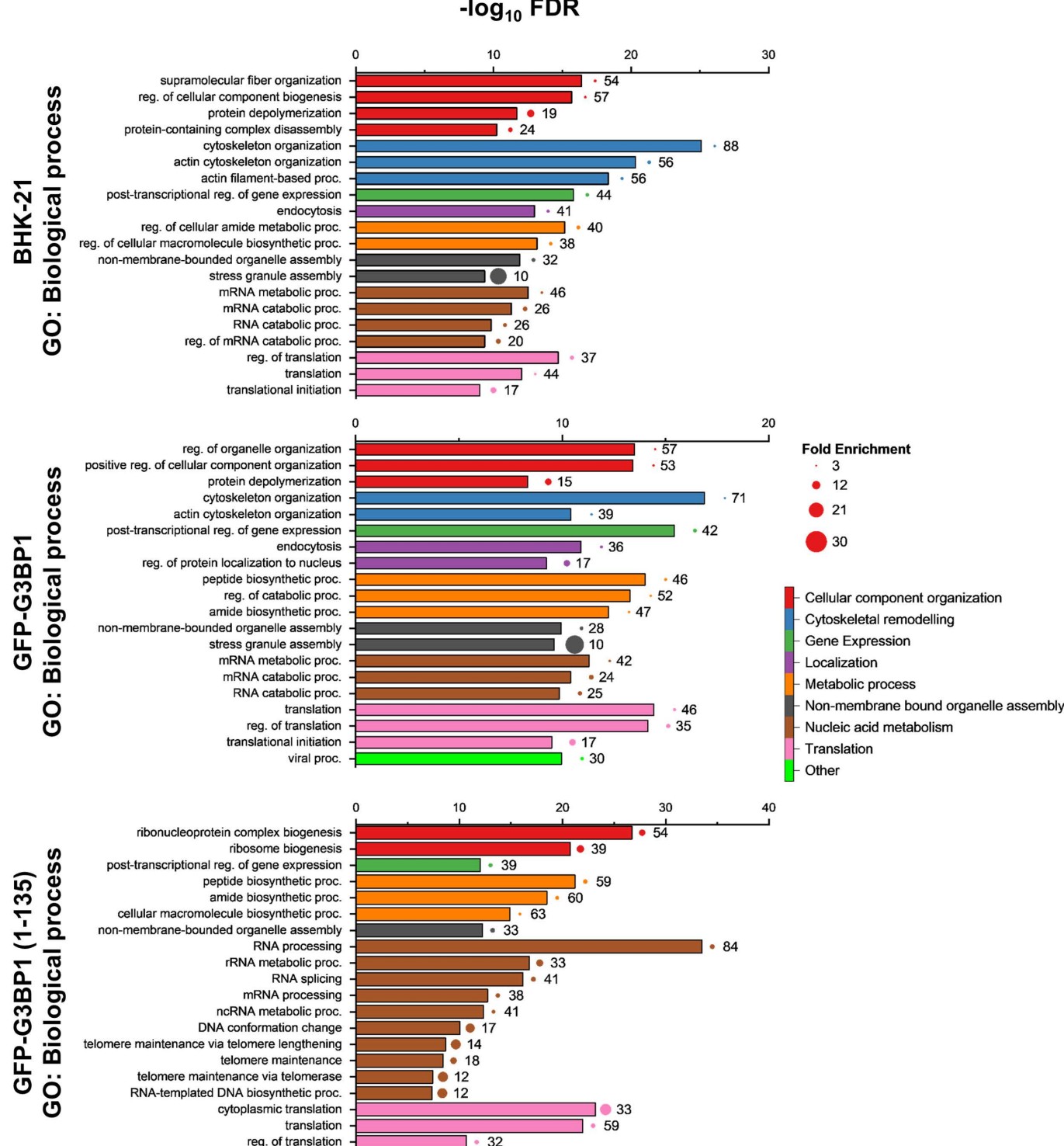

**Fig 4. Gene ontology classification of the nsP3 interactomes based on biological processes.** The top 20 non-redundant pathways are shown, based on their average ranks with their False-discovery rate (FDR) < 10⁻⁶ using ShinyGO v.0.80. The -log₁₀P values indicate their FDRs and the size of the bubble besides the bar indicates the respective fold enrichment. The number beside the bar indicates the total number of proteins identified in the proteome belonging in the corresponding biological process. The different colors represent the clustering of biological processes to a higher node based on the family tree of biological processes.

reporter virus, SFV-Rluc, a reporter SFV expressing *Renilla* luciferase as part of the nonstructural polyprotein (Fig 5A). BIN1 (amphiphysin-2), prominent in the current dataset, was used as a positive control (highlighted in green), as it has been shown to be important for alphavirus replication [24]. Eight interactors detected in this dataset caused >50% reduction in relative luminescence with p-values <0.01 (Fig 5A, highlighted in purple). The cytotoxicity of the transient siRNA knockdown was assessed by measuring the ATP levels, and apart from the knockdown of XAB2, which gave ~60% ATP concentration of the control, depletion of none of the other genes led to a reduction in cell viability (>75% cells viable) when compared with the siRNA control (S3 Fig).

Of the eight proviral factors (excluding BIN1), two (PIK3R2 and DDX3X) have been previously described in connection with alphaviruses, and all the eight have rather different known functions (S2 Table). eIF4G1 is a canonical initiation factor in the cap-dependent translation initiation complex [45], and will be further analyzed below. PIK3R2 is the regulatory subunit of the phosphoinositide-3-kinase signaling pathway, which has been demonstrated previously to be important for alphavirus replication [46,47]. XAB2 is an exonuclease in the nucleus and is involved in stress dependent DNA damage response [48]. Its presence in the proximity of cytoplasmic nsP3 suggests possible recruitment through nsP2, similar to hnRNPs found in the dataset. Another hypothesis may be the natural shuttling of this factor between the nucleus and cytoplasm caused by virus infection [36,38,49]. The possible functions of XAB2 during infection remain to be investigated, especially since siRNA silencing it induced a noticeable cytotoxic effect. The scaffolding protein ANKRD17 is essential for mammalian development, but its molecular functions remain poorly characterized. It could have a role in activating RIG-1-like receptors mediating innate immunity and antiviral activity, as shown for influenza virus infection [50]. However, in the current transient silencing experiment, ANKRD17 had a proviral rather than an antiviral role for SFV. SH3GL1, also known as endophilin A2 is an essential regulator of endocytosis [51]. The SFV replication spherules first arise at the plasma membrane and subsequently undergo endocytosis [31], but endocytosis could also have other roles for virus replication. The other proviral factors will be considered further in the discussion. Analysis of the top interactors with the STRING database using human reference reveals clustering in two distinct sets. They support virus replication either through recruitment of cellular transcription and translation factors in part via the stress granules (Fig 5B, cluster I) or through membrane and cytoskeletal remodeling (Fig 5B, cluster II), which could be involved in the formation of the replication complex and spherules.

We found two novel antiviral factors SF3B3 and eRF3A, for which the siRNA treatment significantly (p<0.01) increased SFV replication. SF3B3 is part of the spliceosome complex, playing important roles in transcription, splicing and DNA repair. Recently, the entire SF3B subcomplex was found to be antiviral for SINV and SFV [44], in agreement with our current results. Interestingly, two major translation factors showed opposing effects on SFV replication, with the peptide release factor eRF3A playing an antiviral role, whereas the initiation factor eIF4G1 was proviral in this experiment.

All factors reducing SFV replication to ≤ 60% and the two antiviral factors were tested for their effects on CHIKV replication. In these experiments, the positive control was the combined knock-down of G3BP1 and G3BP2, which is known to drastically reduce CHIKV replication [23] (Fig 5C). Interestingly, almost all tested siRNAs caused a reduction of CHIKV replication significantly (*p*<0.01), including siRNAs targeting the two factors that increased SFV replication (eRF3A and SF3B3). All the siRNAs that reduced SFV replication, also strongly reduced CHIKV replication, except for SH3GL1 (Fig 5C). The reduction in the case of CHIKV was typically more pronounced, reinforcing the notion that compared to SFV, CHIKV tends to be more dependent on host factors during its replication [19]. The stronger

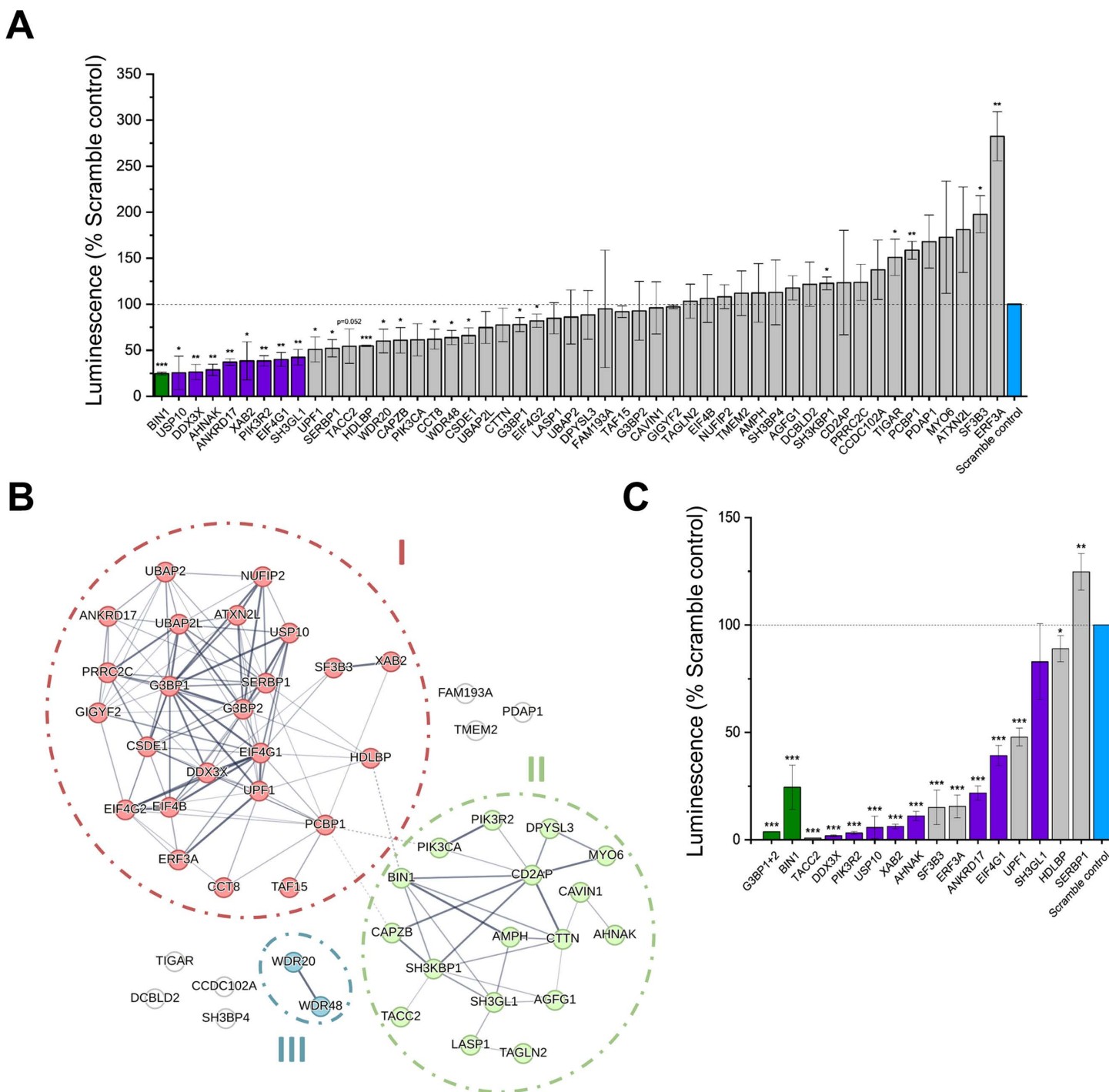

**Fig 5. Effect of transient siRNA mediated knockdown on SFV and CHIKV replication. A)** A pool of four siRNAs against the indicated target was reverse transfected at a concentration of 20 nM in U2OS cells. 70-72 h post transfection the cells were infected with SFV-Rluc at MOI of 0.01 and the *Renilla* luciferase activity was measured at 18 h post infection. The reporter activity in cells treated with siRNA targeting the host factors was normalized to that in cells treated with the non-targeting scramble control siRNA (blue). The graph is a mean of three independent biological replicates and the error bars indicate standard deviation. The *p-values* were calculated using independent Student's t-test; * p<0.05, ** p<0.01 and ***p<0.001. **B)** STRING analysis (with default settings) using human reference database, indicating the clustering of the top interactors in nodes related to recruitment of cellular transcription and translation factors in part via the stress granules (cluster I), and structural functions and membrane remodeling and cytoskeletal rearrangement (cluster II). **C)** Pools of four siRNAs were transfected as in (A), followed by infection with CHIKV-Nluc at MOI of 0.1 and measurement of the reporter activity at 24 h post infection. The columns indicated in purple are the eight host factors that showed >50% reduction in the replication of SFV-Rluc. The data was analyzed as in (A).

effect observed for CHIKV should not be simply due to the slower replication of CHIKV, as the multiplicities of infection have been adjusted to obtain a similar stage of replication for the two viruses (see Materials and Methods). The strongest effect on CHIKV replication was provided by knockdown of TACC2 (which did not reach statistical significance for SFV), a protein regulating microtubule dynamics [52]. SERBP1 and HDLBP (also known as vigilin) did not significantly affect CHIKV replication, in accordance with an earlier study (Fig 5C) [53].

## The impact of translation factors on alphavirus replication

Translation is a crucial process during virus infection, and translation factors have been prominently implicated as proviral proteins for many viruses [54]. Interestingly, eRF3A was recently shown to be a major proviral interactor of Lassa virus and Ebola virus polymerases [55,56]. Translation machinery is recruited to the vicinity of alphavirus replication sites partly through G3BP1 [29]. Therefore, we used small-molecule inhibitors to analyze the effect of the two translation factors affecting SFV replication, the proviral eIF4G1 and the antiviral eRF3A. CC-90009 (also known as Eragidomide) targets eRF3A (eukaryotic peptide chain release factor GTP-binding subunit, also known as GSPT1) for degradation and 4E1RCat inhibits cap-dependent translation by preventing the interaction between eIF4E and eIF4G [57,58]. As expected, CC-90009 caused considerable reduction in the amount of eRF3A (S4A and B Fig), yet there was no observable effect on SFV replication in the presence of CC-90009 (Fig 6A), which also did not exhibit any cytotoxicity (S4C Fig). siRNA treatment for eRF3A (see above) caused an even stronger reduction in the amount of the protein than CC-90009 (S4D and S4E Fig). This may in part explain why a proviral effect was seen with the siRNA but not with the inhibitor, although additional effects of the two treatments can contribute to these outcomes.

In contrast, 4E1RCat significantly reduced SFV replication (Fig 6B) (p<0.001 for concentrations ≥ 6.6 µM). The selectivity index (SI) for 4E1RCat was 4.8 based on the calculated $EC_{50}$=8.0 µM and $CC_{50}$=38.1 µM from the dose-response curves (Fig 6B). The analysis of the expression of SFV non-structural proteins in the presence of 4E1RCat revealed a steady decline in nsP1, nsP2, and nsP3 levels with increasing concentration of the inhibitor (Fig 6C), also indicating reduced virus replication. To assess overall translation during 4E1RCat treatment, U2OS cells were labeled with puromycin (Fig 6D). Since alphavirus infection results in strong inhibition of host translation (compare infected vs. uninfected cells), it was observed that treatment with moderate concentrations of 4E1RCat partially restored normal translation, presumably due to the inhibition of virus replication (Fig 6D).

## Discussion

Here, we have elucidated the proximity interactome of alphavirus nsP3 by genetically fusing the biotin ligase mTB to SFV nsP3. This allowed experiments in multiple cell lines and likely detection of both direct and indirect interactors. Mass spectrometry analysis of the biotinylated proteins revealed that the micro-environment of nsP3 included all the SFV non-structural proteins, consistent with their presence at the replication spherules [6,16]. Notably, the three previously identified and studied direct nsP3 interactors amphiphysin-2 (BIN1), CD2AP, and G3BP1 [24,43,59] were detected as the most prominent host proteins in terms of their fold change enrichment (Fig 3). The dataset is also in agreement with the earlier studies of the nsP3 interactome through co-immunoprecipitation [20,60]. These similarities serve to validate the proximity labeling approach, which provides greater sensitivity, as we identified novel host factors that have not been previously associated with alphaviruses (S2 Table).

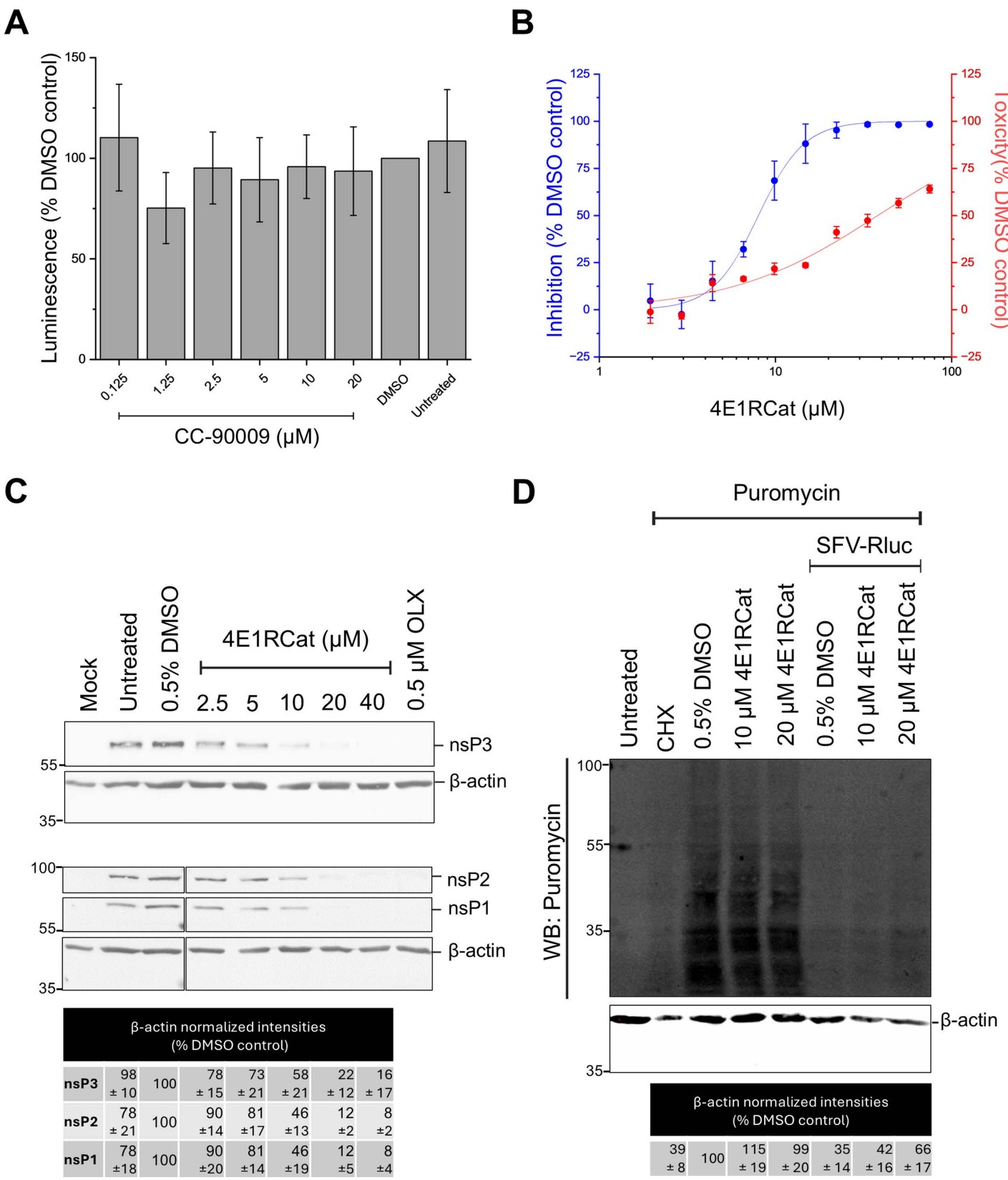

**Fig 6. Effects of translation factor inhibitors on SFV replication and protein translation. A)** The replication of SFV-Rluc in U2OS cells in the presence of the indicated concentrations of Eragidomide (CC-90009), analyzed by the activity of *Renilla* luciferase, normalized to that of DMSO treated infected control cells.

**B)** The dose-response curves of 4E1RCat for SFV-Rluc replication (left scale) and cytotoxicity (right scale) in U2OS cells. The graphs show the average of three independent experiments carried out in quadruplicate, with the error bars representing standard deviation. **C)** Expression of SFV non-structural proteins in cells treated with 4E1RCat. U2OS cells were infected with SFV-Rluc at MOI 0.01 for 18 h in the presence or absence of 4E1RCat, as indicated. Cell lysates were analyzed by Western blotting for nsP1, nsP2, and nsP3; β-actin was used as a loading control. The antiviral obatoclax (OLX) [85] was used as an independent inhibition control. The Western blot shown is a representative image from one of three independent experiments, the numerical values for nsP expression levels are mean of the three experiments ± standard deviation. **D)** Protein translation assay. U2OS cells were incubated with 10 μM or 20 μM 4E1RCat or 0.5% DMSO, and infected with SFV-Rluc at MOI 0.01 for 18 h. The positive control was incubated with 200 μg/mL of cycloheximide (CHX) for 3 h. Then, the cells were exposed to puromycin at 10 μg/mL for 90 min. The lysates were collected, and the total proteins were analyzed by Western blotting using antibodies against puromycin; β-actin was used as a loading control. The numerical values for overall translation are mean of four independent experiments ± standard deviation.

Beyond the replication spherules, nsP3 is found in alternative foci induced when nsP3 breaks down the stress granules induced by infection [27,28]. As shown recently, the nsP3 foci formed during later stages of infection consist of nsP3-multimers in tubule-like structures [16]. Although the tubules co-localize with the viral capsid protein, viral genomic RNA and certain host proteins [16], it is unclear whether they affect encapsidation. Interestingly, no viral structural proteins were detected near nsP3 in our dataset. This may be due to our study focusing on the early stage of viral replication, when nsP3 foci are still largely absent. Secondly, in the geometry of nsP3 tubules the tail region of nsP3, which in nsP3-mTB contains the mTB enzyme, points outward from the tubule surface [16], so that proteins located in the interior of the tubule might remain inaccessible for biotin labeling.

To assess the impact of the identified proteins on alphavirus replication, we performed siRNA-mediated knockdown in cells prior to infection with reporter viruses. We identified three previously known (BIN1, DDX3X, and PIK3R2) and six novel proviral factors, USP10, AHNAK, eIF4G1, SH3GL1, XAB2 and ANKRD17. Of these, AHNAK was only recently linked with virus replication, as it was identified to be a major factor promoting influenza virus and SARS-CoV-2 infection [61]. AHNAK is an enormous ~700 kDa scaffolding protein involved in multiple activities including calcium signaling and coordination of membrane and cyto-skeleton architecture [62]. Since the interaction partner of AHNAK during influenza virus infection is the viral nuclear export protein, it was hypothesized that AHNAK could function in RNA export or virus budding [61]. It is possible that AHNAK's function during alphavi-rus infection is entirely distinct from that during influenza virus infection, and it could be involved in e.g. membrane or cytoskeletal rearrangements. We have shown that AHNAK's proviral roles extend beyond respiratory virus infections, requiring detailed investigation.

The nsP3 interactors can be classified in broad functional categories, including RNA metabolism, stabilization and transcription, protein chaperones, endocytosis, ubiquitin based proteasomal degradation pathways, actin and cytoskeletal remodeling, as well as translation (Fig 4). Nucleic acid metabolic processes are likely associated with viral RNA stabilization as well as with replication/transcription-related processes, as previously described for XRN1 and DDX3X in SINV and VEEV infections [35,39,63]. DDX3X is a helicase with specific roles related to the initiation of translation and has been previously reported as an important host factor for New World alphaviruses [39]. DDX3X has also been indicated as a proviral effector in flaviviruses, picornaviruses, norovirus, herpes simplex virus and human immunodeficiency virus [40]. The silencing of DDX3X is also responsible for the reduction of antiviral responses, which could increase viral replication. Alternatively, or in addition, hijacking of this protein with multi-faceted functions may have a proviral effect due to its impact on RNA metabolism and translation initiation complexes.

Another major group of the interactors are stress granule-associated proteins, which may be present in the nsP3 dataset due to indirect interactions via G3BPs. Stress granule proteins and other RNA-binding proteins were found to be prominent interactors of SFV nsP3, when the protein was expressed alone and immunoprecipitated, but most proteins

present in the current datasets were not detected by that method [64]. Stress granules act as non-membranous cytoplasmic foci that are induced to combat the strain caused by viral infections [65]. The hijacking of the crucial stress granule initiator G3BP1 by nsP3 serves to prevent the formation of stress granules [27,28]. Comparison of the nsP3 interactome in the GFP-G3BP1(1-135) cell line, defective in stress granule formation, allowed us to exclude the majority of stress granule-associated proteins (Fig 3C, S1 S2 Tables). However, some stress granule-related proteins, including UBAP2L and PRRC2C were still present (albeit at a lower fold change) in the dataset of GFP-G3BP1(1-135) cells. In the case of UBAP2L, its interaction with G3BP1 is mediated through RNA [66]. Another major stress granule factor, USP10, was still present among nsP3 interactors in GFP-G3BP1(1-135) cells. USP10 is a G3BP-binding protein that inhibits stress granule formation and interacts primarily with the NTF2 domains of G3BP1 [32]. Notably, USP10 was identified here as strongly proviral for CHIKV and SFV. This suggests that USP10 is the main mediator of proviral effects among the stress granule proteins. As another stress granule-inhibiting protein, USP10 could potentiate the effects of nsP3 in dismantling the stress granules, but the specific downstream effects of USP10 in the context of infection remain to be characterized. Interestingly, USP10 is also proviral for SARS-CoV-2 [67]. These examples suggest that several different viruses may be converging on specific key proteins to enhance their replication.

Recently, the proximity labeling proteomics approach identified eRF3A as a crucial factor in the replication of Lassa fever virus and Ebola virus [55,56]. Interestingly, our experiments showed that eRF3A had an antiviral effect, as siRNA against it increased SFV replication (Fig 5A). Secondly, the compound CC-90009, which induces the degradation of eRF3A, had no effect on SFV replication (Fig 6A), whereas it strongly impaired Lassa fever virus and Ebola virus. This highlights the complex interactions of viruses with the core translational machinery and indicates that different parts of the translational apparatus can provide antiviral effects. This is also apparent in the multiple ways viruses seek to manipulate the translational components by proteolysis, binding interactions, and other means [45]. One recent example is the importance of eIF4G1-containing complexes for influenza virus infection, through interaction with the viral NS1 protein [68].

eIF4G1 was one of the host factors showing a proviral effect during SFV infection. This large scaffolding protein is the central component of the cap-dependent translation complex eIF4F, which additionally contains the cap recognition protein eIF4E and the ATP-dependent helicase eIF4A [45]. The depletion of eIF4G1 should prevent the formation of the eIF4F complex. To further verify the proviral role of eIF4F, we tested the inhibitor 4E1RCat, which blocks the interaction between eIF4E and eIF4G [57]. 4E1RCat inhibited SFV replication in a dose-dependent manner with an $EC_{50}$ of 8.0 µM (Fig 6B) and restored host translation at concentrations sufficient for inhibiting virus replication (Fig 6D). This suggests that translation of the viral non-structural proteins is more stringently dependent on the classical cap-dependent initiation pathway than overall cellular translation. 4E1RCat was also shown to inhibit coronavirus replication [69], raising the exciting possibility that targeting the eIF4F complex might be a strategy for developing broadly acting antivirals. However, 4E1RCat has so far only been used under cell culture conditions, so any potential toxic effects in vivo remain to be studied. Similarly, other proviral proteins, such as AHNAK, which supports influenza virus replication [61], could be potential targets for antiviral compounds.

Limitations of the study: Proteomics based methods are biased by protein abundance and characteristics (size, membrane embedding, etc). Although we find the most prominent previously known nsP3 interactors in our datasets, false positives and negatives must be considered, and additional validation is needed especially to proteins close to statistical significance values, which have not been investigated here by siRNA experiments. The specific molecular mechanisms by which the proviral and antiviral factors act remain to be investigated. As interactors

of one of the replicase proteins, the effects may impinge on the RNA replication stage of the viral life cycle, but they certainly can have other effects.

## Materials and methods

### Cells

BHK-21 (baby hamster kidney; ATCC #CCL-10), U2OS (human osteosarcoma; ATCC #HTB-96), and U2OS-derived ΔΔG3BP1/2 knock-out cells expressing either GFP-G3BP1 WT or GFP-G3BP1 (1-135) mutants [32] (gifts from Gerald McInerney, Karolinska Institutet, Sweden), all were cultured in Dulbecco's modified essential medium (DMEM), supplemented with 10% heat-inactivated fetal bovine serum (iFBS) (Gibco, qualified), 2 mM L-glutamine, 100 U/mL penicillin, and 100 µg/mL streptomycin (Gibco). For the U2OS-derived mutant cell lines, Geneticin (G418) at a final concentration of 0.5 µg/mL was added. Cell lines were maintained in a humidified incubator at 37 °C with 5% $CO_2$. The mutant cell lines (ΔΔG3B-P1/2-U2OS)+GFP-G3BP1-WT and (ΔΔG3BP1/2-U2OS)+GFP-G3BP1(1-135) are referred to as GFP-G3BP1 and GFP-G3BP1(1-135), respectively.

### Plasmids and viruses

To create pCMV-SFV4-nsP3-mTB cDNA clone, the mTB sequence was amplified from a synthetic plasmid containing the mTB sequence (S1 Data) with the linker amino acids GGSGGS at the N-terminus and GGGSGG at the C-terminus. The forward primer 5′-TAATA*CTCGAG*GGCGGATCGGGTGGTTCTATCCCGC-3′ and reverse primer 5′-TATAT*CTCGAG*TCCTCCAGAGCCTCCGCCCTTTTCG-3′ were used for amplification.

The fragment between XbaI and SacI restriction sites containing nsP3 region was digested from pCMV-SFV4 [70] and inserted into pUC18 plasmid. The mTB PCR product was inserted into the pUC18-nsP3_fragment using the unique XhoI site within the SFV nsP3 sequence [71]. After obtaining the nsP3-mTB fusion product, the fragment between XbaI and SacI was transferred back to pCMV-SFV4, resulting in pCMV-SFV4-nsP3-mTB construct. The inserted fragment of mTB was confirmed through sequencing (Eurofins, Germany). In CHIKV-Nluc, the NanoLuc luciferase marker was inserted in the C-terminal domain of nsP3, in the same way as has been previously described for *Renilla* luciferase, using the same LR2006 OPY1 CHIKV strain belonging to the East/Central/South African genotype as basis [72]. The viral constructs will be made available upon request.

To generate the viruses, BHK-21 cells were transfected with infectious cDNA clones of SFV4 wild-type [70], SFV4-nsP3-mTB (see above) and SFV-Rluc [73] using Lipofectamine LTX reagent with Plus reagent (Invitrogen) as per the manufacturer's protocol. To generate CHIKV-Nluc, the plasmid was first linearized with NotI restriction enzyme, and transcribed *in vitro* using mMessage mMachine SP6 transcription kit (Thermo Fisher Scientific). BHK-21 cells were transfected using obtained infectious RNA transcripts and Lipofectamine 2000 reagent (Invitrogen) as per the manufacturer's protocol.

The viruses were collected from the supernatant at 18 h (SFV) or 24 h (CHIKV) post transfection and quantified with plaque titration on BHK-21 cells. The stocks were further amplified by infecting BHK-21 cells at MOI 0.01 for 16 h (SFV) or at MOI 0.1 for 24 h (CHIKV), and the collected viral stocks were quantified with plaque titration. All experiments with CHIKV-Nluc (including virus rescue) were conducted in the Core Facility for Biosafety (ABSL3) of the Institute of Technology, University of Tartu (Ravila 14b, Tartu, Estonia), under the necessary safety approvals.

For plaque titration, the viral stocks were serially diluted in infection medium (minimum essential medium (MEM)/0.2% bovine serum albumin (BSA)/50 mM HEPES/2mM L-gutamine).

The viruses were adsorbed for 1 h at 37 °C on BHK-21 cells. For SFV, the infected cells were covered with MEM-overlay medium (0.8% carboxymethyl cellulose, 3.75% heat-inactivated FBS, 15 mM HEPES pH 7.2, 2.4 mM L-glutamine, 1.2 U/mL penicillin, 1.2 g/mL streptomy-cin,) and incubated at 37 °C for 46-48 h. The overlay medium was aspirated, the cells washed with phosphate buffered saline (PBS) and stained with crystal violet (2% w/v) for 20-30 min at room temperature, and the plaques were counted. For CHIKV, the infected cells were incu-bated for 24 h and subjected to immunoplaque assay as described [74], using rabbit polyclonal CHIKV anti-capsid serum as the primary antibody.

## Biotinylation assay

Cells were infected with SFV4-nsP3-mTB or wild-type SFV4 at the indicated MOI and the viruses adsorbed for 1 h at 37°C. Then, the cells were washed and supplemented with fresh medium. At the indicated times post infection, the medium was replaced with a medium containing 50 μM biotin (Sigma-Aldrich). The biotinylation was stopped at appropriate time points by washing the cells with ice-cold 1× PBS and cell pellets were dissolved in sodium dodecyl sulfate (SDS) sample buffer and boiled for 3 min. The samples were separated in SDS-polyacrylamide gels and analyzed with Western blotting as described below.

## Mass spectrometry analysis of biotinylated proteins

Four 150 mm confluent culture plates of BHK-21 cells were infected with either SFV4-nsP3-mTB or SFV4 at MOI 50. At 2.5 h post infection, the cells were supplemented with a medium containing 50 μM biotin for 15 min. Similarly, four 150 mm confluent plates of GFP-G3BP1(1–135) or GFP-G3BP1 cells were infected with SFV4-nsP3-mTB at MOI 100 for 8 h or 5 h, respectively, or wild-type SFV4 at MOI 100 for 5 h or 4 h, respectively. At the indicated times post infection, the cells were supplemented with 50 μM biotin for 15 min.

Post labelling, cells were washed twice with ice-cold PBS, detached using a cell scraper (Biofil), and collected in chilled falcon tubes. The cells were pelleted at $510 \times g$ for 5 min at 4°C and lysed in ice-cold lysis buffer containing 50 mM Tris-HCl pH 7.4, 500 mM NaCl, 0.2% (w/v) SDS, 1 mM DTT, 2% Triton X-100, protease inhibitor (Pierce Protease mini-inhibitor tablet) and benzonase nuclease (Chem Cruz). The lysates were incubated on ice for 20 min and intermittently vortexed. The lysate was sonicated for 20 pulses (0.3 s ON, 1 s OFF) three times at 30% amplitude (Branson Sonifier SFX150) and diluted 1:1 with 50 mM Tris-HCl pH 7.4. The lysates were centrifuged at $12000 \times g$ for 10 min at 4°C. The samples were incubated with Dynabeads MyOne Streptavidin C1 (Invitrogen) overnight at 4°C with gentle agitation [54].

The beads were separated with DynaMag-2 Magnet (Invitrogen) and washed twice with buffer 1 (2% SDS, 50 mM Tris-HCl), once with buffer 2 (0.1% sodium deoxycholate, 1% triton X-100, 1 mM EDTA, 500 mM NaCl, 50 mM HEPES pH 7.4), once with buffer 3 (0.5% sodium deoxycholate, 0.5% NP-40, 1 mM EDTA, 250 mM LiCl, 10 mM Tris-HCl pH 7.4) and thrice with 50 mM Tris-HCl containing 250 mM NaCl. During the last wash, 20% of the sample was collected separately for validation with SDS-PAGE and the remainder was analyzed using mass spectrometry.

For mass spectrometry sample preparation, beads were washed two times with 100 μl of 100 mM ammonium bicarbonate (Sigma Aldrich). Then 40 μl of ammonium bicarbonate and 1 μl of 0.5 μg/μl trypsin (Promega) was pipetted on to the beads, mixed by vortexing and the mixture was incubated for 2 h at 37 °C with agitation. After incubation, approximately 40 μl of supernatant was collected with a magnetic rack to separate tube, and sample volume was brought to 100 μl by adding 60 μl of 100 mM ammonium bicarbonate. 11.1 μl of tris (2-carboxyethyl) phosphine (final conc. in solution 5 mM; Thermo Scientific) was added to

each sample, and samples were incubated for 20 min at 37 °C with agitation. For alkylation, 12.3 µl of iodoacetamide (final concentration 10 mM; Acros Organics) was added and samples were further incubated for 20 min at room temperature in the dark. pH of the samples was monitored to be 8, and 2 µl of trypsin (same concentration as above) was added and samples were incubated at 37 °C with agitation for 16 h.

After digestion peptides were quenched with 10% trifluoroacetic acid (TFA, VWR), and desalted with BioPureSPN MINI C18 columns (#HUM S18V, Nest Group, USA) following the manufacturer's protocol. The dried peptides were reconstituted in 30 µl Buffer A (0.1% (vol/vol) TFA and 1% (vol/vol) acetonitrile (VWR)) in HPLC water (Fisher Scientific). 1 µl samples were further diluted with 19 µl of HPLC water containing 0.1% (v/v) formic acid. The manufacturer's instructions were followed to load into Evotips (EvoSep, Denmark).

The desalted samples loaded to Evotips were analyzed using the Evosep One liquid chromatography system coupled to a hybrid trapped ion mobility quadrupole TOF mass spectrometer (Bruker timsTOF Pro, Bruker Daltonics) [75] via a CaptiveSpray nano-electrospray ion source (Bruker Daltonics). An 8 cm × 150 µm column with 1.5 µm C18 beads (EV1109, Evosep) was used for peptide separation with the 60 samples per day methods (21 min gradient time). Mobile phases A and B were 0.1% formic acid in water and 0.1% formic acid in acetonitrile, respectively. The samples were analyzed with DDA-PASEF-short_gradient_0.5s-cycletime –method [75,76]. We used the Fragpipe analysis platform (version 18.0) with MSFragger (version 3.5) [77,78], and Philosopher (version 4.2.2) [79] for peptide identification using raw (.d) files as input.

The fasta sequence database used to analyze the BHK-21 data was Golden hamster (*Mesocricetus auratus*). In the case of mutant U2OS cell lines, human (*Homo sapiens*) reference was used along with the necessary modification of removing the G3BP2 (Uniprot ID: Q9UN86) protein from the reference and substituting the wild-type G3BP1 (Uniprot ID: Q13283) with either GFP-G3BP1 or GFP-G3BP1(1-135). Along with these databases, the SFV protein sequences (Uniprot ID: P08411.POLN_SFV, P03315.POLS_SFV) with either wild-type nsP3 or nsP3-mTB was added depending on the virus used for infecting the cells.

For the MSFragger analysis, default settings were kept except that the precursor mass tolerance was set from -50 to 50 ppm and the fragment mass tolerance to 20 ppm. Enzyme specificity was set to "strict trypsin" and two missed cleavages were allowed. Isotope error was set to 0/1/2. Peptide length was set from 5 to 50, and peptide mass was set from 200 to 5000 Da. For all samples carbamidomethylation of cysteine residues was used as static modification, and amino terminal acetylation, oxidation of methionine, were used as the dynamic modification. Maximum number of variable modifications per peptide was set to 3. The false discovery rate for protein identification was set to 1%.

## Western blotting and silver staining

The cell lysates were separated using SDS-PAGE with 4% stacking gel and 10% resolving gel and electro-transferred to a 0.45-µm Amersham Protran nitrocellulose membrane. The membrane was blocked with 5% non-fat milk in Tris-buffered saline (TBS) and incubated with primary antibodies diluted in TBS-0.1% tween-20 (TBST) containing 5% non-fat milk for 1 h at room temperature. The membranes were washed with TBST and detected by incubating with secondary antibodies diluted in TBST/5% non-fat milk for 1 h at room temperature. The membranes were imaged using ChemiDoc MP (Biorad) and analyzed using ImageLab software v.6.0. The antibodies used are listed in S4 Table.

For silver staining, the cell lysates were separated with SDS-PAGE as described, and the gels were fixed with 30% ethanol and 0.5% acetic acid for 30 min and washed with 20% ethanol and then water. The gels were sensitized with sodium thiosulphate (0.02% w/v) for 1 min and washed thoroughly with water. The sensitized gels were stained with silver nitrate (0.2% w/v)

in the dark for 30 min and developed with developing solution (3% w/v potassium carbonate, 0.001% w/v sodium thiosulphate, 0.026% v/v formaldehyde) until stain was visible. The development was stopped with a 5% Tris base solution containing 2.5% acetic acid.

### siRNA transfection and measurement of virus replication

Based on data acquired through mass spectrometry analysis, a pool of siRNA against the first 50 high confidence interactors were tested. U2OS cells (20,000 cells/well) were reverse transfected with 2 pmol of siRNA SMARTpool (Dharmacon, Horizon Discovery) using Lipofectamine RNAiMAX (Invitrogen) on a microtiter plate as per the manufacturer's protocol and incubated at 37°C for 70-72 h.

The siRNA silenced cells were infected with luciferase-based reporter viruses, SFV-Rluc or CHIKV-Nluc, for 18 h at MOI 0.01 or 24 h at MOI 0.1, respectively. Rluc activity was measured in Thermo Varioskan LUX multimode plate reader using *Renilla* Luciferase assay kit (Promega) as per the manufacturer's protocol. NanoLuc activity was measured in CLARIOstar microplate reader (BMG Labtech) using the Nano-Glo Luciferase Assay System (Promega).

To examine the potential cytotoxic effects of the knockdowns, the cell viability was determined through quantifying ATP, which indicates the presence of metabolically active cells, using the Cell Titer Glo 2.0 (Promega) and measured using Thermo Varioskan LUX multimode plate reader. All siRNA transfection experiments were carried out in three independent biological replicates.

For visualizing the knockdown effect of eRF3A (GSPT-1) siRNA, U2OS cells (80,000 cells) were reverse transfected with 8 pmol siRNA as described above. The transfected cells were washed with ice-cold PBS at the indicated timepoints and lysed with SDS for Western blotting as described above.

### Inhibitors

The small molecule inhibitors CC-90009 and 4E1RCat were procured from MedChemTronica AB (Sweden) and dissolved in DMSO. U2OS cells were infected with SFV-Rluc at MOI 0.01 for 18 h, in the presence of different dilutions of the inhibitors. The replication of virus was quantified using *Renilla* luciferase assay (Promega) or by analyzing the treated cell lysates through Western blots. The cytotoxicity due to the inhibitor treatment was assayed with Cell Titer Glo 2.0 (Promega). The luminescence measurements were done with Thermo Varioskan LUX multimode plate reader. To verify the degradation of eRF3A in presence of CC-90009, U2OS cells were incubated with the indicated concentrations of the inhibitor for 18 h. The treated cells were washed with ice-cold PBS at the indicated timepoints and lysed with SDS for Western blotting as described above.

### Puromycylation assay

For verifying the effect of 4E1RCat on total cellular translation, U2OS cells were treated with the inhibitor at the indicated concentrations in infected (SFV-Rluc at MOI 0.01 for 18 h) and uninfected cells. As a positive control for inhibition of translation, cells were treated with 200 μg/mL cycloheximide (Sigma Aldrich) for 3 h. Then all then samples were exposed to 10 μg/mL puromycin (Gibco) for 90 min. Then, cells were washed with ice-cold PBS and lysed with SDS for Western blotting as described above.

### Statistical and bioinformatic analyses

The graphs and data prepared were made with Microsoft Excel 2016 and OriginPro 2024. The statistical significance estimated in luciferase, cell viability and Western blot assays was done with independent Student's t-test and the error bars in figures represent standard deviation.

The dose-response curves for inhibitors to calculate the $CC_{50}$ and $EC_{50}$ values were done using the in-built sigmoidal fit function in OriginPro. Image analyses were conducted using ImageLab v6.0 or Fiji ImageJ v1.53.

The mass spectrometry data obtained as MaxLFQ intensities from the MSFragger analysis was $\log_2$-transformed, normalized, and the empty values were imputed as described [80]. The statistical significance was calculated with independent Student's t-test to determine the enriched proteins. The corresponding gene list obtained was analyzed using ShinyGO v.0.8 (http://bioinformatics.sdstate.edu/go/). The gene ontology analysis was done by setting the false discovery rate (FDR) cut-off at $1\times10^{-6}$ and the functional annotation was done using the GO:Biological processes (GO: BP) human database [81]. For analyzing BHK-21 cells, we used human orthologs of the gene identified. The significant pathways were sorted by average ranks (based on FDR and Fold enrichment) and non-redundant top 30 GO: BP terms were used for further analysis (S3 Table). The Venn diagram was created using a web-tool Eulerr (https://eulerr.co/) [82,83]. The protein-protein network analysis was done with STRING (https://string-db.org/; v. 12.0) with default setting using human proteome as reference [84]. The thickness of lines between the proteins represents the confidence of interaction based on pre-existing data. The clusters were generated using k-means clustering with the number of clusters set to 3.

## Supporting Information

**S1 Fig. Schematic of the engineered virus and time scale of nsP3 expression during infection of U2OS derived cell lines. A)** mTB was inserted to the SFV4 genome (bottom), so that it is located approximately at the center of the C-terminal unstructured region of nsP3 (shown as squiggly lines), followed by the known binding sites for amphiphysin (BIN1) and G3BPs (top). **B and C)** GFP-G3BP1 and GFP-G3BP1(1-135) cells were infected with either **(B)** SFV4-nsP3-mTB or **(C)** SFV4 at MOI 100. Cell lysates were collected at the indicated time points, and analyzed for nsP3/nsP3-mTB expression by Western blotting using antibodies against nsP3; β-actin was used as a loading control. (TIF)

**S2 Fig. Volcano plots of nsP3 interactomes in mutant U2OS cell lines. A and B)** Volcano plots represent the distribution of the proteins detected in GFP-G3BP1 cells **(A)** and GFP-G3BP1(1-135) cells **(B)** through proteomic analysis. High confidence interactors were determined as $\log_2$ Fold change > 1 and $-\log_{10}$ adjusted p-value >1.3 (i.e. p-value < 0.05; independent Student's t-test), indicated with the shaded area. The viral non-structural proteins in the dataset are marked with red, the most prominent previously known interactors are marked with green, and the two translation factors are indicated in yellow. (TIF)

**S3 Fig. Cell viability after transfection with siRNA.** The luminescence levels were determined using CellTiter Glo 2.0 (measuring cellular ATP levels) in U2OS cells, and normalized to those treated with non-targeting scramble control siRNA. The graph is an average of three independent biological replicates and the error bars indicating the standard deviation. The *p-values* were calculated using independent Student's t-test; * p<0.05 and ** p<0.01. (TIF)

**S4 Fig. Effects of CC-90009 and siRNA treatment on eRF3A levels and cell viability. A)** U2OS cells were treated with the indicated concentrations of CC-90009 (see Materials and Methods), and cell lysates were analyzed by Western blotting for eRF3A. **B)** Quantitation of eRF3A expression based on three independent experiments. **C)** Cell viability of U2OS cells after CC-90009 treatment. The luminescence levels were determined using CellTiter Glo

2.0 (measuring cellular ATP levels). **D)** U2OS cells were treated with siRNA against eRF3A or with the scramble control, and cell lysates were analyzed by Western blotting for eRF3A. The time points 72 h and 90 h represent the start and end points of the corresponding SFV infection experiments. **E)** Quantification of eRF3A expression based on three independent experiments. The error bars indicate standard deviation. The p-values were calculated using independent Student's t-test; *** $p < 0.001$.
(TIF)

**S1 Table. Proximal proteins identified by mass spectrometry.** Excel files with fold change and p-values of biotin enriched proteins.
(XLSX)

**S2 Table. Functions of the top 50 high-confidence interactors from BHK-21 interactome.** Excel file listing the proteins, their functions, and known interactions with alphavirus replication.
(XLSX)

**S3 Table. Gene ontology of top interactors identified by mass spectrometry.**
(XLSX)

**S4 Table. List of Antibodies used.**
(XLSX)

**S1 Data. MiniTurbo sequence, and raw data used for calculation of graphs.**
(XLSX)

**S2 Data. Mass spectrometry raw data.**
(XLSX)

## Acknowledgments

We thank Gerald McInerney and David Mentrup for critical reading of the manuscript.

## Author contributions

**Conceptualization:** Aditya Thiruvaiyaru, Sari Mattila, Tero Ahola.

**Data curation:** Aditya Thiruvaiyaru, Sari Mattila, Tero Ahola.

**Formal analysis:** Aditya Thiruvaiyaru, Sari Mattila, Tero Ahola.

**Funding acquisition:** Sari Mattila, Tero Ahola.

**Investigation:** Aditya Thiruvaiyaru, Sari Mattila, Mohammadreza Sadeghi, Krystyna Naumenko.

**Methodology:** Aditya Thiruvaiyaru, Sari Mattila, Markku Varjosalo.

**Project administration:** Tero Ahola.

**Supervision:** Andres Merits, Tero Ahola.

**Writing – original draft:** Aditya Thiruvaiyaru, Tero Ahola.

**Writing – review & editing:** Aditya Thiruvaiyaru, Sari Mattila, Mohammadreza Sadeghi, Krystyna Naumenko, Andres Merits, Markku Varjosalo, Tero Ahola.

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
