## [Decision Letter · Decision Letter 0]

14 Jan 2025

PPATHOGENS-D-24-02500

Proximity interactome of alphavirus replicase component nsP3 includes proviral host factors eIF4G and AHNAK

PLOS Pathogens

Dear Dr. Ahola,

Thank you for submitting your manuscript to PLOS Pathogens. After careful consideration, we feel that it has merit but does not fully meet PLOS Pathogens's publication criteria as it currently stands. Therefore, we invite you to submit a revised version of the manuscript that addresses the points raised during the review process.

Please submit your revised manuscript within 60 days. If you will need more time than this to complete your revisions, please reply to this message or contact the journal office at plospathogens@plos.org. Please include the following items when submitting your revised manuscript:

We look forward to receiving your revised manuscript.

Kind regards,

Gorben P Pijlman

Guest Editor

PLOS Pathogens

Sonja Best

Section Editor

PLOS Pathogens

 Sumita Bhaduri-McIntosh

Editor-in-Chief

PLOS Pathogens

orcid.org/0000-0003-2946-9497

Michael Malim

Editor-in-Chief

PLOS Pathogens

orcid.org/0000-0002-7699-2064

**Additional Editor Comments:**

All reviewers were enthusiastic about your work and made helpful suggestions to improve the manuscript. I share their view that the work has an innovative experimental approach and advances the field of alphavirology with regard to the elusive functions of the nsP3 protein.

Rev#2 has made import suggestions to quantify expression levels of antiviral proteins and restructure several Figures with the aim to better support the conclusions. Some additional experimentation may be needed to address these concerns.

**Journal Requirements:**

At this stage, the following Authors/Authors require contributions: Aditya Thiruvaiyaru, Sari Mattila, Mohammadreza Sadeghi, Krystyna Naumenko, Andres Merits, Markku Varjosalo, and Tero Ahola. Please ensure that the full contributions of each author are acknowledged in the "Add/Edit/Remove Authors" section of our submission form.

- ® on page: 26.

- TM on pages: 23, 24, 25, 26, 27, and and 32.

5) We notice that your supplementary Figures are included in the manuscript file. Please remove them and upload them with the file type 'Supporting Information'. Please ensure that each Supporting Information file has a legend listed in the manuscript after the references list.

6) Please ensure that the funders and grant numbers match between the Financial Disclosure field and the Funding Information tab in your submission form. Note that the funders must be provided in the same order in both places as well.

**Reviewers' Comments:**

Reviewer's Responses to Questions

**Part I - Summary**

Reviewer #1: In the manuscript: “Proximity interactome of alphavirus replicase component nsP3 includes proviral host factors eIF4G and AHNAK” the authors describe creating SFV which contains an in frame insertion of biotin ligase miniTurbo within non-structural protein 3. Subsequent infections resulted in the biotinylation of viral and host proteins in the proximity of nsP3. With this strategy the authors identified many host proteins that reside in the proximity of nsP3 during the early phase of SFV replication. Subsequently, with an siRNA approach the role of top interactors in SFV and CHIKV replication was identified. The role of nsP3 in alphavirus replication remains enigmatic, therefore this explorative study and the identified putative virus-host interactions can be a starting point for novel studies focussing on alphavirus replication or the identification of antivirals. I do have a few suggestions that may improve the manuscript

Reviewer #2: In this manuscript, Thiruvaiyaru et al. have identified novel interactors of alphavirus protein nsP3 by applying a proximity labelling methodology within an authentic virus context. Replication of viral RNA is achieved by a complex interplay of the viral non-structural proteins nsP1, nsP2, nsP3, and nsP4 and host factors and takes place within membranous invaginations into the plasma membrane, termed spherules. To build upon the library of known alphavirus replication-associated factors, the authors determined the interactome of Semliki Forest virus (SFV) nsP3 fused to miniTurbo in baby hamster kidney (BHK) and human osteosarcoma cells (U2OS). The top hits were included in a knockdown siRNA screen in U2OS cells, which revealed eight proviral interactors whose knockdown decreased SFV infection by 50% and five antiviral interactors whose knockdown increased SFV infection. The effect on SFV infection of the proviral translation initiation factor eIF4G1 and antiviral translation termination factor eRF3A were assessed by pharmacological inhibition, revealing a strong proviral effect of eIF4G1 whereas manipulation of eRF3A levels had no effect. The authors are the first to report proximity biotinylation within an authentic alphavirus context to determine nsP3 interactors, which expands upon the current knowledge of SFV replication and translation. The manuscript is written clearly and offers a strong theoretical framework. However, the second half of the manuscript contains experiments in which important controls are absent which precludes drawing the authors’ conclusions regarding translation initiation. Furthermore, the choice of placing certain data in the main manuscript and other important data in the supporting information undermines the coherence of the result section. Hence, the manuscript should be reinforced with control experiments and the data should be restructured as outlined below, before it is suitable for publication.

Reviewer #3: Thiruvaiyaru and colleagues describe a proteomics screen for interactors of the alphaviral eiigmantic nsP3 protein using a recombinant biotin-transferase tagged Semliki Forest Virus (SFV) and proximity biotinylation/MS approach. The study is flawless, well-conceived, described, conducted and presented in a very readable and comprehensive narrative. Next to the hand-full of known interactors serving as method validation, the study identifies an exhaustive list of >300 new candidate host factors of which more than 100 are shared between 2 human and 1 hamster cell line, including cells with a functional KO for G3BP1 dependent stress granule (SG) formation. Latter is importance to demonstrate SB independent nsP3 host-interactions. Top hits are functionally validated by siRNA mediated knockdown, identifying two major host protein clusters (cellular transcription/translation machinery, and membrane/cytoskeleton remodeling) essentially involved in SFV (and by extension Chickungunya virus) replication. One thus identified factors, eIF4G1 is confirmed to serve as proviral factor for SFV protein synthesis by selective inhibition of SFV replication and protein synthesis using selective small molecule inhibitor 4E1RCat.

**Part II – Major Issues: Key Experiments Required for Acceptance**

Reviewer #1: (No Response)

Reviewer #2: The authors treat U2OS cells with the compound CC-90009 that targets eRF3A, one of the significant antiviral hits in the siRNA screen, for degradation and do not observe an effect on SFV infection. The authors should show that CC-90009 treatment results in a decrease in eRF3A protein level by western blot and compare this to the decrease in eRF3A protein level realized by siRNA knockdown, as to explain the large difference between results from Fig. 6A and Fig. 7A.

The figures should be condensed into fewer figures and data should be exchanged between the supporting information and the main manuscript to increase coherence. Fig. 1 and 2 should be combined, focusing on the setup of the proximity labelling approach. Fig. 3 is informative, but the limited conclusions drawn from the blots does not warrant its presence in the main manuscript and should be moved to the supporting information. Fig. S4 should be added to Fig. 6 to showcase the broader implication of the identified hits in alphavirus replication. This is especially important because the effect of eRF3A knockdown differs between the SFV and CHIKV. The current data presented in Fig. 7 should be condensed to two panels (see minor comments) and Fig. 8 and Fig. S5 should be added to Fig. 7, so that the mechanistic data about translation are presented collectively.

The protein translation assay presented in Fig. S5 contains important results on the selectivity of the effect of 4EIRCat treatment on viral versus host translation. The authors should add these data to the main manuscript and perform quantitation on the lane intensities of three independent experiments to support the statement in Line 344 that host translation is restored following 4EIRCat treatment.

Reviewer #3: 1. Results are discussed in a comprehensive way. The authors may however need to mention and discuss limitations of their study, including chosen methodology, issues of validation, and lack of further insight in molecular mechanisms and remaining questions at which specific steps in the viral life cycle these host factors may be involved.

2. Gene ontologies (GOs) are used to classify thus identified host factors. The authors should discuss how relevant and reliable such GOs classifications (which are defined based on their physiological role) can be considered seen that viral machinery may hijack them in a non-canonical function.

**Part III – Minor Issues: Editorial and Data Presentation Modifications**

Reviewer #1: Line 61: “which seems to consist of nsP3”. This statement alone is rather vague. Perhaps the authors can briefly add what observations make this likely.

Lines 145-155: It is unclear from the text what functions are restored by the truncated G3BPs and what the replicative fitness of SFV is in G3BP1/2 knockout cells with or without expression of the truncated proteins. Please clearly provide this information either as experimental data or with citations to previous work.

Consider rephrasing lines 148-150: “We used the double knock-out cell lines GFP-G3BP1 and GFP-G3BP1(1-135), reconstituting either the full-length G3BP1, or only the N-terminal NTF2-like domain, each fused with green fluorescent protein” to clarify that double knockout lines were reconstituted with G3BP1, G3BP1 (1-135) or the NTF2-like domain fused to GFP.

Line 169: “Comparing the patterns between GFP-G3BP1 and GFP-G3BP1(1-135) cells”. Do you mean G3BP1/2 knockout cells that express GFP-G3BP1 and GFP-G3BP1(1-135)?

lines 249-250: “We selected the top high-confidence hits from the interactome and studied their effect on SFV infection in U2OS cells”. Please specify selection of this top. Is this referring to the top50 mentioned in line 190?

Line 327: “In contrast, 4E1RCat significantly reduced SFV replication (Fig 7B) in non-toxic concentrations.” Significance is not indicated in text, figure legend or the figure itself.

Fig. 7A and B: There seems to be an error in the presentation of the error bars, vertical lines are missing in some of the bars.

Reviewer #2: The authors should consider adding the individual data points of the independent experiments to their graphs.

• Line 79: specify what is meant by ‘normal’.

• Line 85: add comma after specifically.

• Lines 86-88: add reasoning on what the benefit is of analysing the nsP3 interactome in the absence of stress granules.

• Line 107: why was such a high MOI chosen to analyse viral replication, especially during replication assessment in Fig. 1C? Generally, these assessments are performed at low MOI.

• Fig. 1.

o 1C: Y-axis label should spell titer for consistency.

o Figure legend: state which MOI was used. State what the data points in 1C represent, i.e., mean ± SD of how many independent replicates.

o Consider adding a schematic representation of SFV-nsP3-mTB showing the location of mTB and the different domains of nsP3.

• Fig. 2.

o 2A+B: write the label Streptavidin vertically and add vertical line to increase clarity.

• Line 153: repeat which time point was chosen eventually for the MS analysis in U2OS.

• Fig. 3.

o 3A,B,C: add streptavidin as staining label (similar to Fig. 2).

o Capitalize the P in nsP3 in the lane names.

• Fig. 4.

o 4A: the shaded purple area results in the color-coded data points being difficult to differentiate. Consider removing the purple box as the dotted purple lines with labels are clear already. Be consistent in capitalization and use of spaces in figures (Fold change vs Fold Change; spaces in Fold change label but not in p-value label).

o 4B+C: be consistent with labelling, choose either “not present” or “missing”.

o Line 183: exchange “protein numbers” for “number of proteins”.

• Line 229: add “have *been previously shown” and “*are important”.

• Fig.5.

o Add FDR to axis label with (-log10 P) as unit in between brackets.

o Remove grammar mistakes from figure legend.

• Line 249: add that these were the top hits from the BHK-21 interactome.

• Line 251: add on which location in the viral genome the luciferase is encoded.

• Line 258: replace “the reduction” with “a reduction”.

• Line 297: adjust to “effect on CHIKV replication”.

• Fig. 6.

o Line 309: exchange average for mean.

o Line 310: adjust to “error bars indicate”.

o Line 314: add punctuation.

• Line 322: repeat that eIF4G1 was proviral for SFV and eRF3a was antiviral for SFV to improve clarity.

• Line 323: it is confusing to use GSPT1 and eRF3a interchangeably. Consider referring to eRF3a/GSPT1 from the start of the manuscript.

• Fig. 7.

o Apart from the major comments regarding this figure, mentioned above, the authors should condense the data presented here. It seems that the data in panel B and C are exactly the same. Panel B can be removed and an SD should be added to panel C and D. Panels C and D can be combined using left and right Y-axes. Cell viability data of CC-90009 should be added.

• Line 356: the authors do not show data on the detection of both direct and indirect interactors and should nuance this statement.

• Line 400: add “a” to proviral effect.

• Line 401: it is unclear what is meant by “interactions in RNA metabolism”, please rephrase.

• Line 409: similar to the minor comment about the word “normal” in the introduction, can the authors define what they mean by that.

• Line 415: was USP10 still present in the GFP-G3BP1(1-135) cells in general or in the interactome dataset? Please specify.

• Line 419: what is meant by “USP10 could potentiate the effects of nsP3” please clarify.

• Lines 426-428: see the major comment: if CC-90009 treatment would be in accordance with the siRNA screen, replication would have significantly increased, which is not the case.

• Line 429: “can be modified” is vague, please specify.

• Line 437: add “complex” to end of sentence.

• Line 443: can the authors expand on the use of non-classical translation initiation pathways used by host cells.

• Line 444: is there an indication that compounds targeting eIF4F could be safely used in vivo without causing adverse effects?

• Line 474: add space between LR2006 and OPY1.

• Line 544: typo, *samples.

• Line 636: add space after (B).

• Fig. S1: add that beta-actin was used as loading control to figure legend.

• Fig. S2: see comments regarding Fig. 4A.

• Fig. S3: add that ATP levels were measured to the figure legend.

• Table S4: add the final concentration of the commercial antibodies.

Reviewer #3: 1. Line 14: I think “confirmed” would be more appropriate instead of “identified”.

2. Figures: The authors use different fonts (with and without serifs, font sizes) in graph legends (e.g. Fig.1A,B vs.Fig.4 B ,Dvs. A,C). I suggest to harmonize.

3. In Fig. 4A yellow dots are hard to see.

4. Fig.5 shows fold enrichment using circles which are hard to distinguish seen the small size differences. Again the yellow circle (for SG assembly) is hard to see.

5. Line 270: …since siRNA silencing <could as="" be="" for="" not="" purpose="" this="" used=""> it induced a noticeable …

6. Line 292: discrepancy between CHIKV and SFV restriction and proviral activity needs to be discussed. Can discrespancies e.g. in much higher fold-reduction in CHIKV replication be explained by diffenrces in replication kinetics of both viruses?

7. Fig. 6B. Please increase font size.

8. EC50 and CC50 values for 4E1RCat should be rounded to maximally one digit (8.0 and 38.1µM, or 8 and 38µM) see that they are derived from curve fits and SI will not essentially change.

9. Fig.7: Please harmonize graphs. All graphs should have the same y-axis and should repot the same (relevant) concentration scale. Please consider reporting EC (inhibition) and CC (viability) values in combined graphs. This provides a direct visualization of specific activity (selectivity) versus unspecific.

10. Fig.S5 is a nice control for specific dependency of viral protein expression on cap-dependent translation. However, densitometry of bands/smears should be provided for quantitative support. Also rest of Western blots may benefit from formal quantification.

11. Line 426: “In accordance” is not correct. Use of CC-9009 as chemical probe does not confirm eRF3A as host factor.

12. Table S4: Please provide working dilution for antibodies.</could>

PLOS authors have the option to publish the peer review history of their article (what does this mean? ). If published, this will include your full peer review and any attached files.

**Do you want your identity to be public for this peer review?** For information about this choice, including consent withdrawal, please see our Privacy Policy .

Reviewer #1: No

Reviewer #2: No

Reviewer #3: **Yes: ** Kai Dallmeier

**Figure resubmission:**
---

## [Decision Letter · Decision Letter 1]

17 Mar 2025

Dear Dr. Ahola,

We are pleased to inform you that your manuscript 'Proximity interactome of alphavirus replicase component nsP3 includes proviral host factors eIF4G and AHNAK' has been provisionally accepted for publication in PLOS Pathogens.

Best regards,

Gorben P Pijlman

Guest Editor

PLOS Pathogens

Sonja Best

Section Editor

PLOS Pathogens

Sumita Bhaduri-McIntosh

Editor-in-Chief

PLOS Pathogens

orcid.org/0000-0003-2946-9497

Michael Malim

Editor-in-Chief

PLOS Pathogens

orcid.org/0000-0002-7699-2064

Reviewer Comments (if any, and for reference):

Reviewer's Responses to Questions

**Part I - Summary**

Reviewer #3: Many thanks to the authors for considering and discussing my previously raised concerns. All issues have been addressed appropriately. I hope my interventions helped to enhance the manuscript.

**Part II – Major Issues: Key Experiments Required for Acceptance**

Reviewer #3: (No Response)

**Part III – Minor Issues: Editorial and Data Presentation Modifications**

Reviewer #3: (No Response)

PLOS authors have the option to publish the peer review history of their article (what does this mean? ). If published, this will include your full peer review and any attached files.

**Do you want your identity to be public for this peer review?** For information about this choice, including consent withdrawal, please see our Privacy Policy .

Reviewer #3: **Yes: ** Kai Dallmeier

---

## [Editor Report · Acceptance letter]

Dear Dr. Ahola,

We are delighted to inform you that your manuscript, "Proximity interactome of alphavirus replicase component nsP3 includes proviral host factors eIF4G and AHNAK," has been formally accepted for publication in PLOS Pathogens.

Best regards,

Sumita Bhaduri-McIntosh

Editor-in-Chief

PLOS Pathogens

orcid.org/0000-0003-2946-9497

Michael Malim

Editor-in-Chief

PLOS Pathogens

orcid.org/0000-0002-7699-2064